# Determination of Dehumidification Capacity of Water Wall with Controlled Water Temperature: Experimental Verification under Laboratory Conditions

**Katarina Cakyova** [1,2,*]☺, **Frantisek Vranay** [3], **Marian Vertal** [3] **and Zuzana Vranayova** [3]☺

1. Center for Research and Innovation in Construction, Faculty of Civil Engineering, Technical University of Košice, 042 00 Košice, Slovakia
2. Institute of Building Structures, Faculty of Civil Engineering, Brno University of Technology, 602 00 Brno, Czech Republic
3. Institute of Architectural Engineering, Faculty of Civil Engineering, Technical University of Košice, 042 00 Košice, Slovakia; frantisek.vranay@tuke.sk (F.V.); marian.vertal@tuke.sk (M.V.); zuzana.vranayova@tuke.sk (Z.V.)
* Correspondence: katarina.cakyova@tuke.sk; Tel.: +421-944-731-360

**Abstract:** Water elements with flowing water on the surface are common in buildings as a form of indoor decoration, and they are most often perceived as passive humidifiers. However, by controlling water temperature, they can be also used for air dehumidification. The dehumidification capacity of indoor water elements was investigated experimentally under laboratory conditions. For the experimental verification of dehumidification capacity, a water wall prototype with an effective area of falling water film of 1 m$^2$ and a measuring system were designed and developed. A total of 15 measurements were carried out with air temperatures ranging from 22.1 °C to 32.5 °C and relative humidity from 58.9% to 85.6%. The observed dehumidification capacity varied in the range of 21.99–315.36 g/h for the tested measurements. The results show that the condensation rate is a dynamic process, and the dehumidification capacity of a water wall strongly depends on indoor air parameters (air humidity and temperature). To determine the dehumidification capacity of a water wall for any boundary conditions, the equations were determined based on measured data, and two methods were used: the linear dependence between humidity ratio and condensation rate, and nonlinear surface fitting based on the dependence between the condensation rate, air temperature, and relative humidity.

**Keywords:** water wall; falling water film; relative humidity; temperature; condensation rate; dehumidification

## 1. Introduction

In the field of the research and development of water walls for building applications, water walls where water flows between two solid materials are mainly presented. This is a closed system and is used to adjust thermal comfort, especially air temperature. The first water wall of this type reported in the available literature was the one built in 1947 at the Massachusetts Institute of Technology by Hoyt Hottel and his students with the aim to use a water wall as a passive solar system [1]. It has been proven that a water wall system used as a green building façade can enhance the energy performance [2–5] and also fire protection of buildings [5].

Another group of water walls (Figure 1) represents decorative water features often located in the interior of public spaces such as shopping centers, hospitals, libraries, the entrance areas of office buildings, and hotels, but also airports. In this case, water falls on solid material and is in direct contact with indoor air, so there is an assumption that in addition to thermal comfort, other parameters of the indoor environment can also be affected. Water elements similar to green walls are perceived as a significant aesthetic element of the indoor environment and represent one of the patterns of biophilic design [6–8].

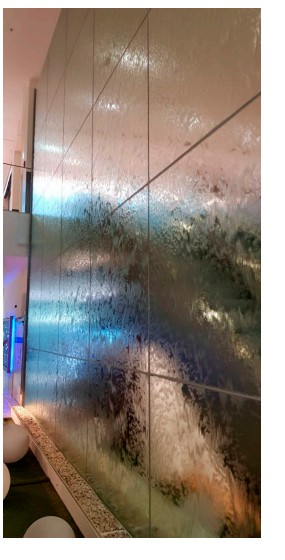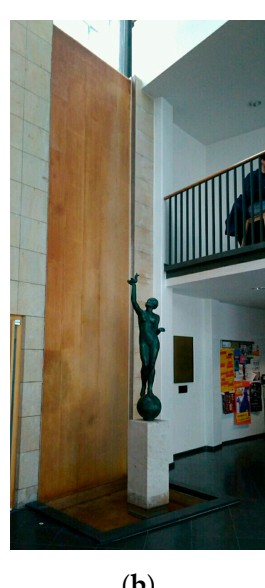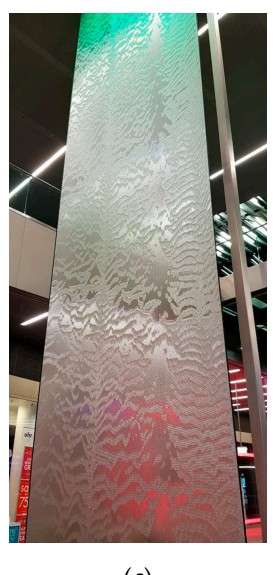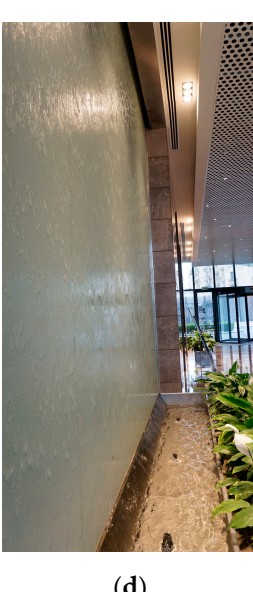

| (**a**) | (**b**) | (**c**) | (**d**) |

**Figure 1.** Water wall examples: (**a**) Shopping center, Vienna, Austria; (**b**) Library, Brno, Czech Republic; (**c**) Railway station, Vienna, Austria; (**d**) Office building, Baku, Azerbaijan.

In the case of indoor greenery (e.g., living walls), many studies have been conducted that define the different systems and designs of these elements [9–12]. It was shown that indoor greenery offers several benefits, such as producing oxygen through photosynthesis [13]; affecting the physical parameters of indoor air quality, mainly temperature and humidity [14–16], $CO_2$ concentration, and acoustics [13,15]; and providing an aesthetically pleasing environment that increases productivity and lowers stress in the building's occupants [17]. The use of indoor plants in some situations could also be a potential problem, however, mainly in the context of humidity and fungal respiratory diseases in patients [10,14].

On the other hand, the issue of indoor water bodies, which also represent the possibility of bringing nature inside a building, has not been explored in detail. There is no clear division that describes these elements, even though water elements are increasingly becoming part of the interior of buildings. Water elements in the interior of a building enhance the experience of a place through the seeing, hearing, or touching of water and are associated with reducing stress, increasing feelings of tranquility, and lowering heart rate and blood pressure, as well as improving concentration and memory restoration [8,18]. It was found that the sound of water has been positively associated with restoration [19], especially in connection with dissatisfaction with environmental noise through windows [20]. However, there are also differences in the way different genders respond to water features, with male subjects tending to respond to water features more frequently than female subjects [18].

At present, water walls where the water circulates are mostly used for an evaporation cooling effect, as a consequence of which the air temperature decreases and water temperature increases [1,21,22]. In addition to temperature, air humidity also changes and increases. This effect may cause an unpleasant level of thermal comfort, especially in hot and humid conditions. Subsequent air treatment and dehumidification may also result in significant energy loads. A study from India showed that direct evaporation cooling was not as effective during the months of July and August, and some dehumidification is required to maintain thermal comfort indoors [23]. Research from Malaysia (which has a hot and humid climate) showed hybrid cooling strategies—thermal stack flue, cross ventilation, and water walls (evaporative cooling)—located in the atriums of office buildings. Due to use of this hybrid system, indoor temperatures were lower than outdoor temperatures, and indoor humidity fluctuated from 68% to 92% during working hours. This system does not increase RH significantly, but this value can still cause some level of discomfort [24]. The high humidity of air can also lead to problems associated with condensation, mainly in

the case of air conditioners without regulation of humidity—for example, radiant cooling systems [25,26]. The evaporative cooling potential of water walls is more suitable for hot and dry climates. However, there is still a risk of Legionella bacteria with increasing water temperature [27]. Additionally, there is an assumption that regulation of water temperature can lead to a useful impact on indoor air temperature and humidity [28]. Research conducted by Fang et al. [29] showed the potential of using water walls with water temperature regulation. However, in this case, the experimental measurements took place in real conditions without control of the physical parameters of the inner microclimate in a specific environment of sorption-active elements. Thus, the resulting effect of the water element on the environment is closely related to the boundary conditions for the given environment; additionally, only the change in individual physical parameters was described, while the share of the water wall for this change was not clearly quantified.

At present, many studies deal with energy consumption in buildings, especially in the context of reaching the required temperature [30–32]. However, achieving the required humidity level is also important. Today, around 10–15% of the total energy consumed by a building is used to achieve the required humidity level [33]. Moreover, based on climate change and the global warming effect, there is an assumption that this percentage will increase several times in the coming years [34,35]. Similarly, as water changes the hydrothermal behavior of buildings with green components [36], the presence of water elements in interior spaces can affect the parameters of the indoor microclimate. This forces us to effectively utilize all elements and to learn more about the potential of individual elements in the building.

The presented paper fills the gap of knowledge in the field of effective use of water walls with controlled water temperature, with a focus on the dehumidification capacity of water walls that is defined by condensation rate under varying boundary conditions. The novelty of the contribution lies mainly in the fact that the water wall and its dehumidification potential are determined in laboratory conditions at various boundary conditions without sorption-active materials. The main object is the quantification of water wall dehumidification capacity via equations of condensation rate that can be used for different spaces of buildings. The paper connects the aesthetic element (water wall) with the functional use (dehumidification) in the interior of buildings. Quantification of dehumidification capacity of water walls with controlled water temperature as a device through an equation is helpful for its design, operation, and implementation in any space.

## 2. Materials and Methods

### 2.1. Research Scheme

Experimental verification under laboratory conditions was chosen to quantify the dehumidification potential of water walls. In Figure 2, it is possible to see a schematic illustration of the methodology of the work. Firstly, the assumption of water wall use was defined. The second step was to design and create a full-scale prototype of a water wall, and then the laboratory conditions were determined and specified. The next step was to perform measurements under different boundary conditions and analysis of the obtained data where condensation rate is expressed by incensement of weight condensate in time. Finally, the co-dependence of parameters (air temperature and humidity) and condensation rate was evaluated using two methods—linear and nonlinear analysis. The result is a comparison of both methods and the definition of water walls with controlled water temperature as a device with an impact on the humidity of the indoor environment. The Microsoft Excel 2016 and OriginPro 9.0 programs were used for the analysis and presentation of the results.

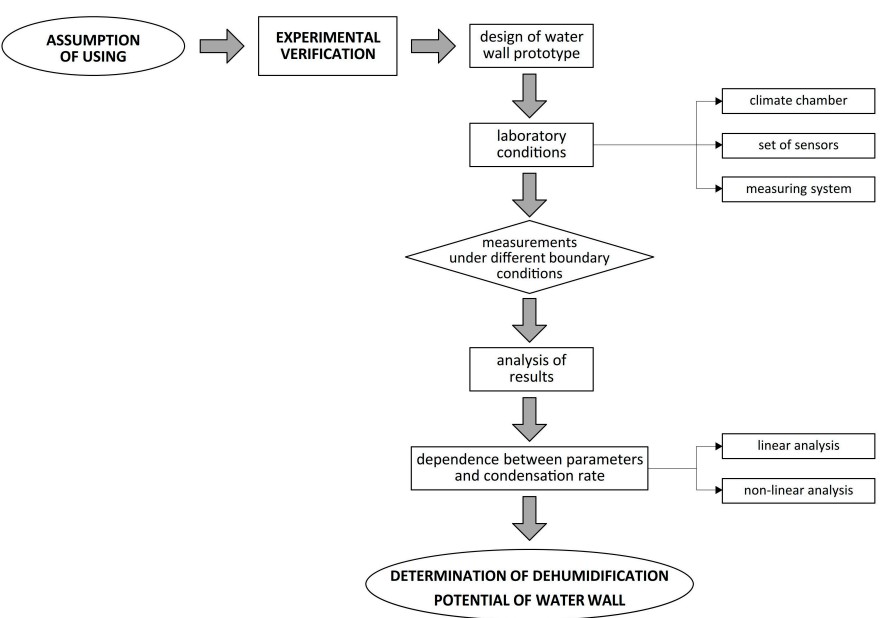

**Figure 2.** Research scheme.

### 2.2. Assumption of Water Wall Use to Dehumidify Air

For the effective use of water walls with a controlled water temperature, the phase change between liquid and gas associated with condensation was used. If water vapor comes into contact with a surface whose temperature is below the dew point of the air, the condensation process occurs, and the vapor changes phase and becomes liquid. In the present paper, the surface with a temperature below the dew point that was used is the falling water film with an average temperature of 14 °C. This process decreases the content of water vapor in the air.

In addition to mass transfer, heat transfer also occurs. Considering the problem of conjugated heat and mass transfer, three basic elements of the system can be specified: the solid body (glass), the thin falling water film flowing down the glass, and the surroundings of the gas mixture (air) [37]. The study of film flow is difficult because the flow itself exhibits large uncertainties and instabilities. Such large uncertainties can be attributed to its diverse, complex, and irregular wave structures [38]. At the top of a falling film unit, i.e., at the liquid inlet, the film is smooth. As the film flows downwards, there is a simultaneous development of hydrodynamic profiles [39]. During the flow of the water film, a characteristic wave pattern is created, which breaks into wave segments with different sizes and shapes. The wave pattern also depends on the flow velocity of the surrounding air [40]. Many studies [37,39,41,42] have proven that film thickness is one of the most essential parameters playing a vital role in determining heat and mass transfer performance. The mass flow rate, but also the shape, surface roughness, or porosity of the solid material after which it flows, have a significant influence on the formation and thickness of the water film and its energy benefit [43,44]. The problem of the falling water film is most often studied theoretically via numerical analysis [37] or via an experimental investigation [42].

### 2.3. Design of Water Wall Prototype

There are various designs and construction methods of decorative water features. In this paper, the construction method based on an overflow edge was used. For the experimental verification, the full-scale water wall prototype was designed (Figure 3). The prototype consists of bottom and upper water tanks with rectangular shapes made of polypropylene material (thickness 6 mm) using a method based on fusion welding; this ensured the tanks were watertight. From the bottom tank, the water is pumped and supplied to the upper water tank, and subsequently, the water spills over the overflow

edge, then flows down the glass and forms a continuous falling water film on it. The effective area of the water film is 1 m$^2$; the dimensions are 710 mm x 1420 mm, and thus, the width to height ratio is 1:2.

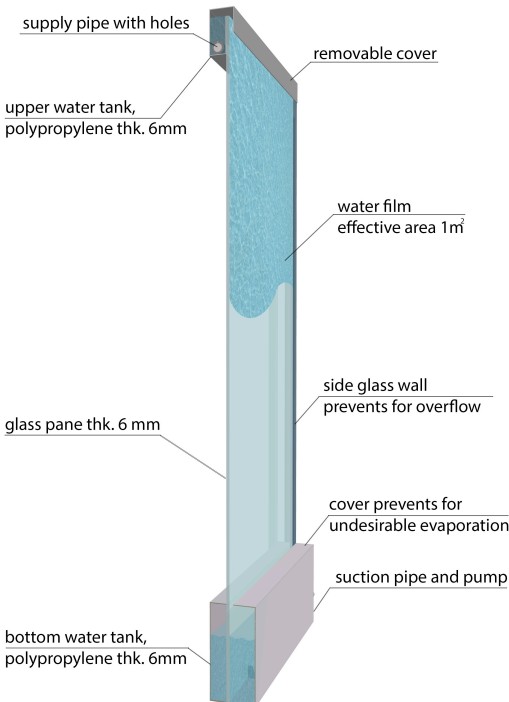

**Figure 3.** Design of water wall prototype.

The formation and thickness of the water film affect, among other things, the geometry of the upper part of the water wall (Figure 4). Into the collecting tank (1) is inserted a perforated pipe (2) for the supply of water (8). The length of the perforated pipe (2) is 700 mm, and the diameter of the holes is 3 mm, with an axial distance of 10 mm (Figure 5a). The orientation of perforation is downwards, which is essential for a smooth inlet of water to the tank and for forming a water film along the entire width of the water wall. The glass pane (4) forms the overflow edge (9) and is supplemented by protruding parts at its side edges to prevent undesired water overflow. A silicone adhesive (7) was used to connect the collecting tank (1) and glass pane (4). A ball valve (10) fastened to the collecting tank (1) serves to empty it if necessary. The collecting tank (1) is anchored to the supporting metal structure (5) by screws (6) and is supplemented by a cover (3). A cover is used also for the bottom tank to prevent unwanted evaporation. The solution of the upper part of the water wall is part of utility model PUV 50129-2017 no. 8710 [45].

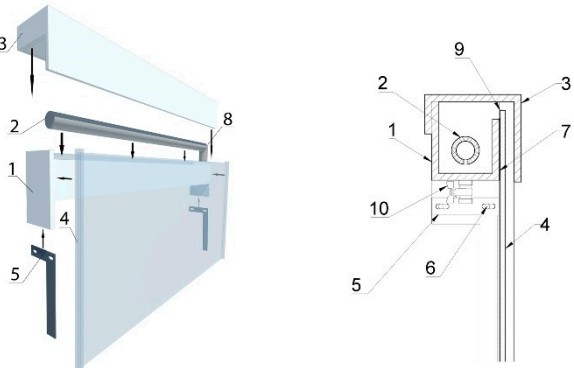

**Figure 4.** Design of water wall prototype.

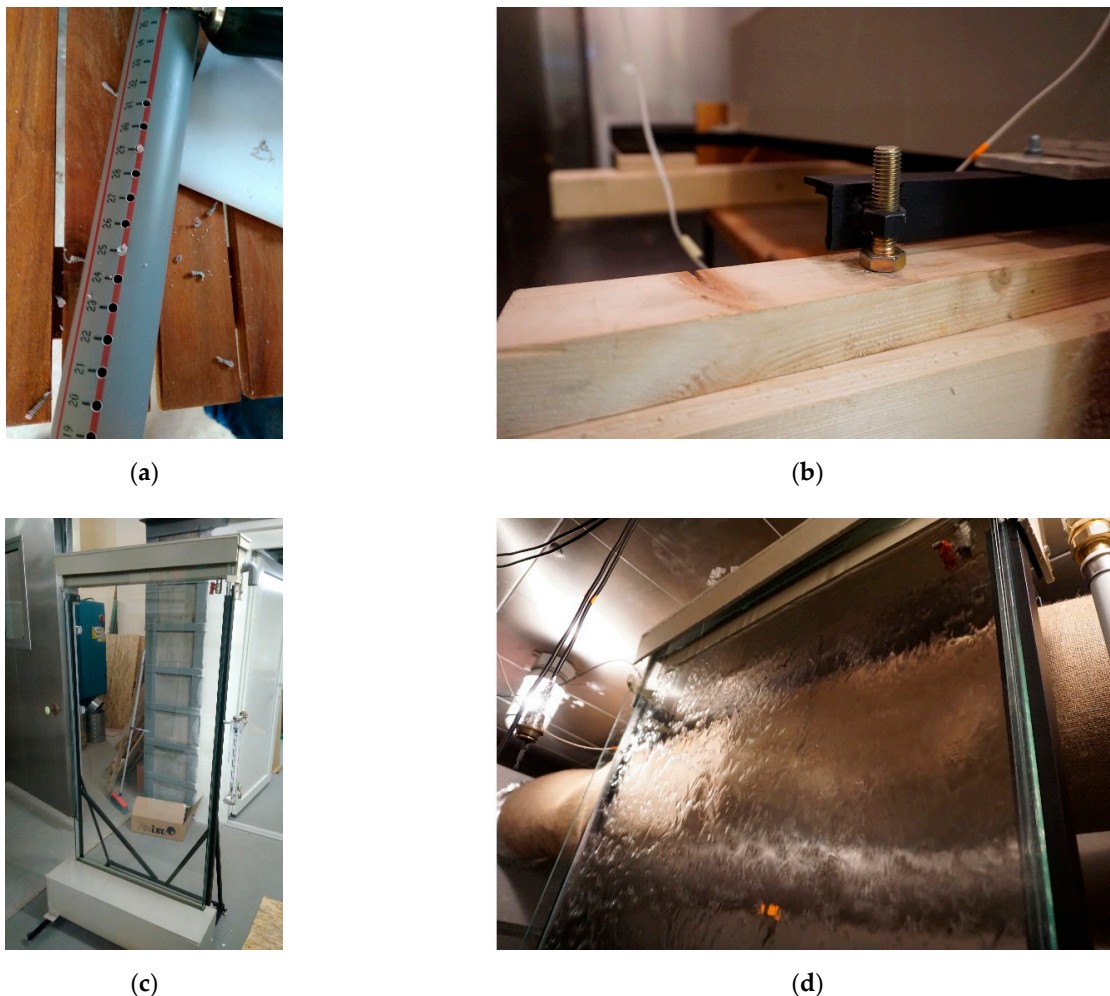

**Figure 5.** Parts of water wall prototype: (**a**) Perforated pipe; (**b**) Screws for the horizontal position; (**c**) Final water wall prototype; (**d**) Water film created on the glass pane.

Furthermore, to form a water film along the entire width of the water wall, the overflow edge must be in a perfectly horizontal position. For this purpose, there are 4 screws located in the lower part of the metal structure (Figure 5).

### 2.4. Laboratory Conditions

#### 2.4.1. Climate Chamber

The water wall prototype was tested under laboratory conditions using a climate chamber (Figure 6) located in the laboratory of the Faculty of Civil Engineering of the Technical University of Košice, where it is possible to set steady boundary conditions (air temperature and humidity can be controlled). The chamber is fully closed, airtight, and consists of two separate rooms. The air temperature and humidity can be controlled for each room separately. For the experiment, just one room was used, with inner dimensions of $3.95 \times 1.60 \times 2.85$ m and volume of air of $18.01$ m$^3$. The temperature ranged from $-20$ °C to $+125$ °C, and the RH of the chamber was adjustable from 20% to 95%. The climatic chamber was chosen to avoid mass exchange between the chamber's walls, which are made of stainless steel, and air. Due to this, the moisture buffer effect of materials can be ignored, and the clean dehumidification capacity of the water wall can be determined. The climate chamber was adjusted and adapted for measurement purposes. The modification of the chamber involved the creation of a textile tunnel that was deployed to the air supply. This intervention contributed to a uniform air distribution in the chamber and, at the same time, avoided a heavy draft.

### 2.4.2. Set of Sensors

In the chamber was installed a set of measuring sensors, which recorded boundary conditions throughout all the experiments. To measure the water temperature, the Ahlborn NTC sensor FN 0001 K was used. Two sensors measured the water temperature in the bottom tank and two in the upper tank. Moreover, two sensors type Ahlborn FHAD 46-C0 were installed to measure the temperature and the relative humidity of the air in front of the water film, and one behind the glass. The limiting deviations of the used sensors are described in Table 1.

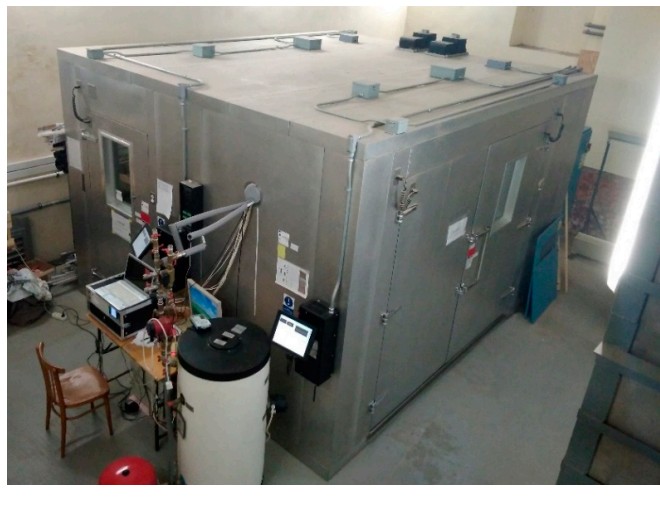

(**a**)

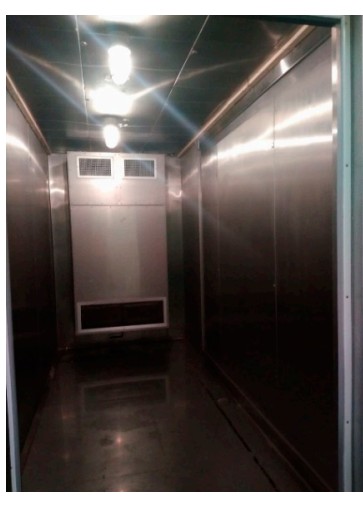

(**b**)

**Figure 6.** Climate chamber: (**a**) Exterior view; (**b**) Interior view.

**Table 1.** Specifications of measuring sensors.

| | Parameter | Device | Range | Deviation |
|---|---|---|---|---|
| Water | temperature | NTC sensor FN 0001 K | 0 to 70 °C | ±0.2 K |
| Air | humidity | FHAD 46-C0 | 10 to 90% RH | ±2.0% RH at nominal temperature (+23 °C) |
| | temperature | | 5 to 60 °C | typical ±0.2 K (maximum ±0.4 K) |

All the sensors were calibrated before use and were connected to a control unit (type Ahlborn) for the storage of the measured parameters. The AMR Win Control software was used to set up experiments and collect the measured data. This ensured the continuous recording of the measured values.

### 2.4.3. Measuring Systems

The water wall prototype was adapted for experimental testing in accordance with the scheme (Figure 7). The prototype was placed on an elevated wooden construction, the bottom water tank was connected with a measuring cylinder, and the different types of components, i.e., valves, flow meter, and plate heat exchanger, were added to the system.

Because there is no standard method for measuring the condensation rate [46], a system based on the weighting of condensate increment measurement was developed (Figure 8). For this purpose, the Radwag laboratory scale connected to a measuring cylinder was used. Its weight limit is 6000 g to an accuracy of 0.01 g. Another important restriction is the fact that ambient relative humidity should not exceed 85%. Same as the sensors, the laboratory scale was connected to the Ahlborn control unit. To achieve the right conditions for condensation, the plate heat exchanger was added to the system and connected to well water. Due to this, the water temperature was constant with a value of around 14 °C. After the start of the experiment, the individual parts of the water wall are flooded, and water needs to be added to the level of overflow created in the measuring cylinder. Subsequently, water from the bottom water tank flows into the measuring cylinder. From

it, water is supplied into the upper part of the water wall by the pump and cooled by the plate exchanger. During the experiment, the water increments caused by the condensation process on the falling water film overflow to the plastic collecting tank, which was put on a laboratory scale. The flow rate velocity is controlled by the shut-off valve, and for accurate monitoring, the flow meter was used. Throughout all the experiments, the flow rate was 450 l/h, which was determined to be the optimal flow rate during past experiments [47]. Via this system, it is possible to accurately and continuously record the condensation rate (condensate formed on the water film) over time.

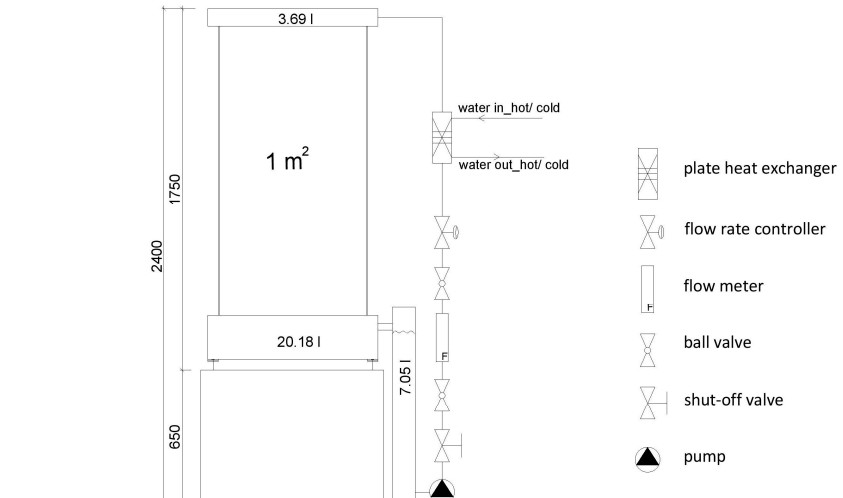

**Figure 7.** Schematic diagram of basic setup of the measurement system.

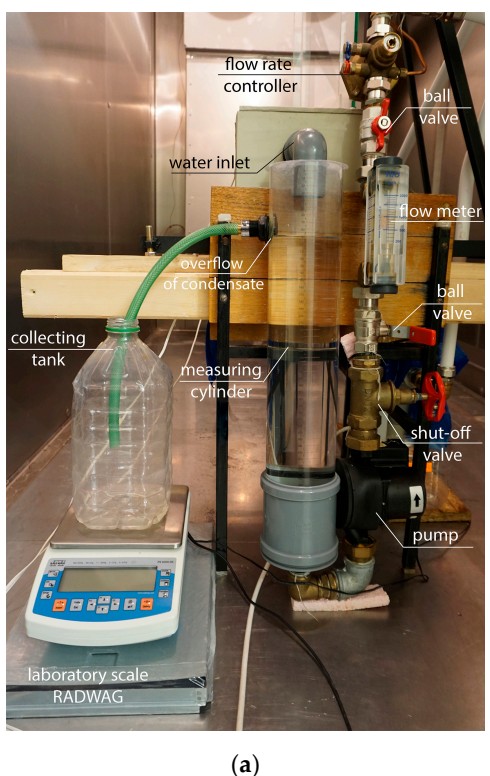

(**a**)

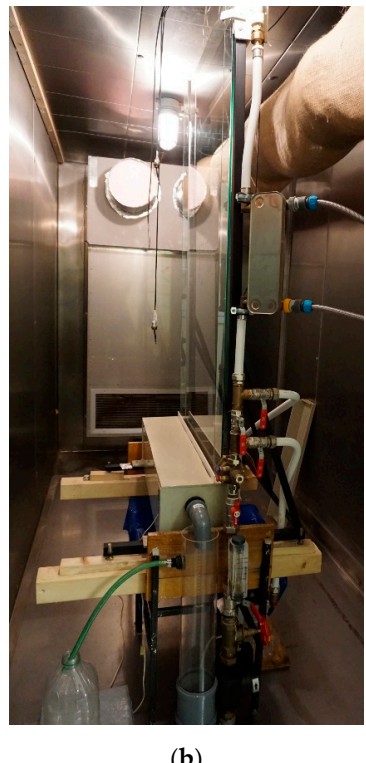

(**b**)

**Figure 8.** The measuring system for dehumidification performance: (**a**) Detailed view of the system; (**b**) Overall view of the system and the water wall located in the climate chamber.

## 3. Results and Discussion

A total of 15 measurements were carried out, with 2 repetitions to determine the dehumidification capacity of the water wall. During the measurements, air temperature varied from 22.1 °C to 32.5 °C, and relative humidity changed from 58.9% to 85.6% (Figure 9).

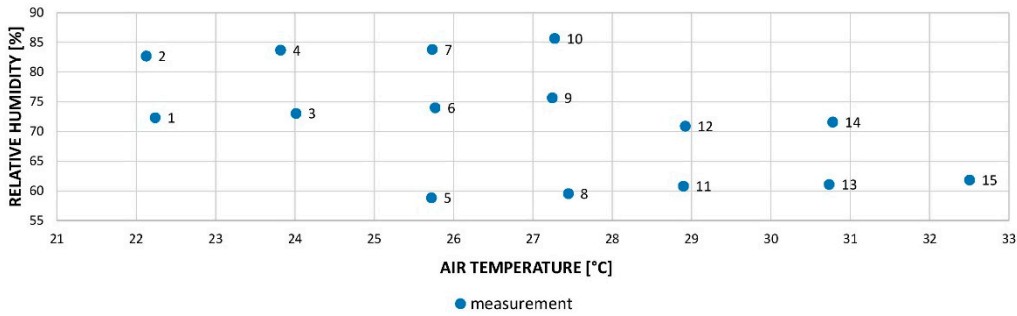

**Figure 9.** Scatter plot of boundary conditions during measurements.

The recording of measurements was performed in a 5 min time step. To avoid exceeding the measuring limit of the laboratory scale, the duration of the experiments ranged from 7 to 15 h. The obtained data from sensors were analyzed and evaluated to average values for each measurement. This means that an average water temperature, average air temperature, and average relative humidity were determined. The average water temperature was 14.8 °C ± 0.3 in the bottom tank and 14.0 °C ± 0.2 in the upper tank. To determine the condensation rate, for each measurement, the weight of condensate in time was evaluated. The resulting values of condensation performance of the water wall prototype and boundary conditions are evaluated in Table 2. This table also shows that the condition for condensation has been met: the water temperature must be lower than the dew point temperature of the air. The dew point temperature ($T_{dp}$) was calculated using the measured boundary conditions of air according to the calculation formula, as shown in Equation (1).

$$T_{dp} = \frac{243.5 \times \ln\left(\frac{RH}{100} \exp \frac{17.67 \times T}{243.5 + T}\right)}{17.67 - \ln\left(\frac{RH}{100} \exp \frac{17.67 \times T}{243.5 + T}\right)} \tag{1}$$

By comparing two groups of measurements, it is possible to observe that the condensation capacity of a water wall depends on the temperature and relative humidity of the air. The first group includes measurements 5, 6, and 7, in which the air temperature was approximately equal (25.7 °C). The second group comprises measurements 8, 9, and 10, when the air temperature was approximately 27.3 °C, which is 1.6 °C higher than that of the first group. In the first case, the air temperature was 25.7 °C, and three measurements were carried out with relative humidity of 58.9% (RH_1), 74.0% (RH_2), and 83.8% (RH_3). In the second case, the air temperature was 27.3 °C and relative humidity 59.5% (RH_1), 75.7% (RH_2), and 83.6% (RH_3). The increase in relative humidity was similar in both cases. The plots (Figure 10) represent the courses of condensate weight created on the water film depending on time.

In the first case (Figure 10a), when relative humidity was 58.9%, the hourly gain of condensate was 21.99 g/h; at a relative humidity of 74.0%, the hourly increase in condensate was 148.81 g/h; and when relative humidity was 83.8%, the condensation performance was 236.73 g/h.

In the second case (Figure 10b), when the relative humidity was 59.5%, the hourly gain of condensate was 68.81 g/h; at a relative humidity of 75.7%, the hourly increase in condensate was 218.19 g/h; and when relative humidity was 85.6%, the condensation performance was 315.36 g/h.

The measurement results show that when the air temperature was 25.7 °C and relative humidity 58.9%, the condensation performance was 21.99 g/h. An increase in relative

humidity of 15.1% represents a rise in condensate weight by 126.82 g, which is 6.8 times higher, whereas with an increase of 24.9%, condensate weight increases by 214.74 g, which is 10.8 times higher compared to when relative humidity is 58.9%.

When the air temperature increased by 1.6 °C to 27.3 °C, the dehumidification performance for a relative humidity of 59.5% was 68.81 g/h. An increase in relative humidity of 16.2% represents a rise in condensate weight by 149.38 g, which is 3.2 times higher, whereas with an increase of 26.1%, condensate weight increases by 246.55 g, which is 4.6 times higher compared to when relative humidity is 59.5%.

The results (Table 3) show a co-dependence of the dehumidification capacity of the water wall and boundary conditions. Even though the increase in relative humidity is comparable in both groups, the dehumidification capacity of the water wall is incomparable.

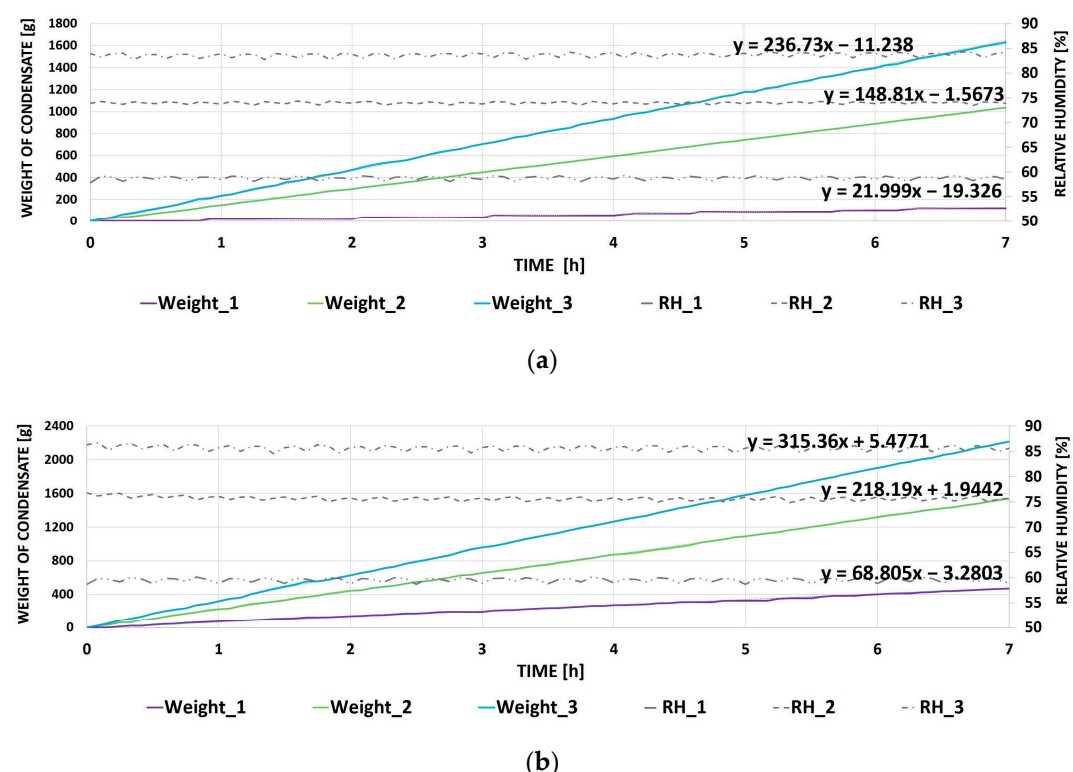

**Figure 10.** Courses of condensate weight increment: (**a**) First case—air temperature of 25.7 °C; (**b**) Second case—air temperature of 27.3 °C.

### 3.1. Analysis of Results Using Linear Dependence

Air humidity can be expressed in various ways. One of them is the humidity ratio. Based on the measured boundary conditions of air (relative humidity and air temperature), the humidity ratio was determined (Table 4). The plot in Figure 11 shows the linear dependence between humidity ratio (g/kg) and condensate rate (g/h).

From the graphical solution of linear dependence, Equation (2) was derived:

$$CR = 32.982 \times \chi - 376.64 \tag{2}$$

where CR is the condensation rate (g/h) and $\chi$ is the humidity ratio (g/kg). The correlation coefficient ($R^2$) of the equation is 0.9589. With the equation, it is possible to determine the water wall's condensation capacity for any humidity ratio.

In the next step, the condensation rate was calculated using Equation (2), and the percentage of deviation compared to the measured values was specified. This comparison, together with the boundary conditions, is shown in Table 4.

**Table 2.** Resulting values of measured parameters.

| | Measurement | | | | | | | | | | | | | | |
|---|---|---|---|---|---|---|---|---|---|---|---|---|---|---|---|
| | **1** | **2** | **3** | **4** | **5** | **6** | **7** | **8** | **9** | **10** | **11** | **12** | **13** | **14** | **15** |
| Temperature (°C) | 22.2 ± 0.1 | 22.1 ± 0.1 | 24.0 ± 0.1 | 23.8 ± 0.1 | 25.7 ± 0.1 | 25.8 ± 0.1 | 25.7 ± 0.2 | 27.5 ± 0.1 | 27.2 ± 0.1 | 27.3 ± 0.2 | 28.9 ± 0.2 | 28.9 ± 0.1 | 30.7 ± 0.1 | 30.8 ± 0.1 | 32.5 ± 0.1 |
| Max Temperature (°C) | 22.4 | 22.4 | 24.2 | 24.0 | 26.0 | 26.0 | 26.1 | 27.7 | 27.5 | 27.7 | 29.3 | 29.3 | 31.0 | 31.1 | 32.8 |
| Min Temperature (°C) | 21.9 | 22.0 | 23.9 | 23.6 | 24.6 | 25.6 | 25.3 | 27.2 | 26.9 | 26.7 | 28.5 | 28.6 | 30.5 | 30.4 | 32.0 |
| Relative humidity (%) | 72.3 ± 1.4 | 82.7 ± 1.0 | 73.1 ± 1.4 | 83.7 ± 1.0 | 58.9 ± 1.4 | 74.0 ± 1.2 | 83.8 ± 1.3 | 59.5 ± 1.4 | 75.7 ± 1.0 | 85.6 ± 0.6 | 60.8 ± 0.9 | 70.9 ± 0.9 | 61.1 ± 1.0 | 71.6 ± 0.9 | 61.9 ± 0.9 |
| Max Relative humidity (%) | 75.1 | 84.5 | 75.5 | 85.5 | 62.7 | 76.2 | 86.3 | 62.1 | 78.0 | 87.1 | 62.5 | 72.7 | 63.0 | 73.5 | 64.5 |
| Min Relative humidity (%) | 70.2 | 80.2 | 71.0 | 81.3 | 56.8 | 72.4 | 81.5 | 57.3 | 73.6 | 84.1 | 59.2 | 69.4 | 59.6 | 69.9 | 60.4 |
| Dew point temperature (°C) | 17.02 | 19.04 | 18.88 | 20.89 | 17.07 | 20.78 | 22.78 | 18.86 | 22.56 | 24.65 | 20.56 | 23.1 | 22.36 | 25.04 | 24.23 |
| Water temperature_upper (°C) | 13.6 ± 0.02 | 14.0 ± 0.01 | 13.7 ± 0.02 | 14.0 ± 0.02 | 13.9 ± 0.02 | 13.9 ± 0.05 | 14.0 ± 0.02 | 13.8 ± 0.02 | 14.2 ± 0.01 | 14.3 ± 0.01 | 14.1 ± 0.01 | 14.2 ± 0.03 | 14.2 ± 0.01 | 14.3 ± 0.01 | 14.3 ± 0.01 |
| Water temperature_down (°C) | 14.1 ± 0.03 | 14.4 ± 0.01 | 14.4 ± 0.06 | 14.6 ± 0.02 | 14.6 ± 0.03 | 14.6 ± 0.03 | 14.8 ± 0.02 | 14.4 ± 0.01 | 15.0 ± 0.02 | 15.2 ± 0.02 | 14.8 ± 0.02 | 15.1 ± 0.04 | 15.1 ± 0.02 | 15.4 ± 0.02 | 15.3 ± 0.02 |
| Duration of experiment (h) | 15.0 | 15.0 | 12.0 | 15.0 | 12.0 | 12.0 | 14.0 | 12.0 | 7.0 | 7.0 | 15.0 | 14.0 | 15.0 | 7.0 | 7.0 |
| Condensation rate (g/h) | 46.02 | 104.29 | 88.95 | 173.42 | 21.99 | 148.81 | 236.73 | 68.81 | 218.19 | 315.36 | 118.62 | 219.92 | 177.4 | 298.17 | 240.4 |

**Table 3.** The resulting comparison and evaluation of two selected groups of measurements.

| | Air Temperature (°C) | Relative Humidity (%) | Condensation Rate (g/h) | Increscent (-) |
|---|---|---|---|---|
| First group of measurements | 25.7 | 58.9 | 21.99 | reference value |
| | | 74.0 | 148.81 | 6.8 times higher than reference value |
| | | 83.8 | 236.73 | 10.8 times higher than reference value |
| Second group of measurements | 27.3 | 59.5 | 68.81 | reference value |
| | | 75.7 | 218.19 | 3.2 times higher than reference value |
| | | 85.6 | 315.36 | 4.6 times higher than reference value |

**Table 4.** Comparison of measured and calculated condensation rate using humidity ratio.

| | Measurement | | | | | | | | | | | | | | |
|---|---|---|---|---|---|---|---|---|---|---|---|---|---|---|---|
| | **1** | **2** | **3** | **4** | **5** | **6** | **7** | **8** | **9** | **10** | **11** | **12** | **13** | **14** | **15** |
| Temperature (°C) | 22.2 | 22.1 | 24.0 | 23.8 | 25.7 | 25.8 | 25.73 | 27.5 | 27.2 | 27.3 | 28.9 | 28.9 | 30.7 | 30.8 | 32.5 |
| Relative humidity (%) | 72.3 | 82.7 | 73.1 | 83.7 | 58.9 | 74.0 | 83.8 | 59.5 | 75.7 | 85.6 | 60.8 | 70.9 | 61.1 | 71.6 | 61.9 |
| Humidity ratio (g/kg) | 12.45 | 14.18 | 14.03 | 15.94 | 12.49 | 15.83 | 17.94 | 14.02 | 17.7 | 20.15 | 15.61 | 18.31 | 17.48 | 20.64 | 19.63 |
| Measurement: CR (g/h) | 46.02 | 104.29 | 88.95 | 173.42 | 21.99 | 148.81 | 236.73 | 68.81 | 218.19 | 315.36 | 118.62 | 219.92 | 177.4 | 298.17 | 240.4 |
| Equation: CR (g/h) | 33.99 | 91.04 | 86.10 | 149.09 | 35.31 | 145.47 | 215.06 | 85.77 | 207.14 | 287.95 | 138.21 | 227.26 | 199.89 | 304.11 | 270.80 |
| Absolute difference | 12.04 | 13.25 | 2.85 | 24.33 | 13.32 | 3.34 | 21.67 | 16.96 | 11.05 | 27.41 | 19.59 | 7.34 | 22.49 | 5.94 | 30.40 |
| Percent deviation (%) | 26.16 | 12.70 | 3.20 | 14.03 | 60.55 | 2.25 | 9.16 | 24.65 | 5.06 | 8.69 | 16.51 | 3.34 | 12.67 | 1.99 | 12.64 |

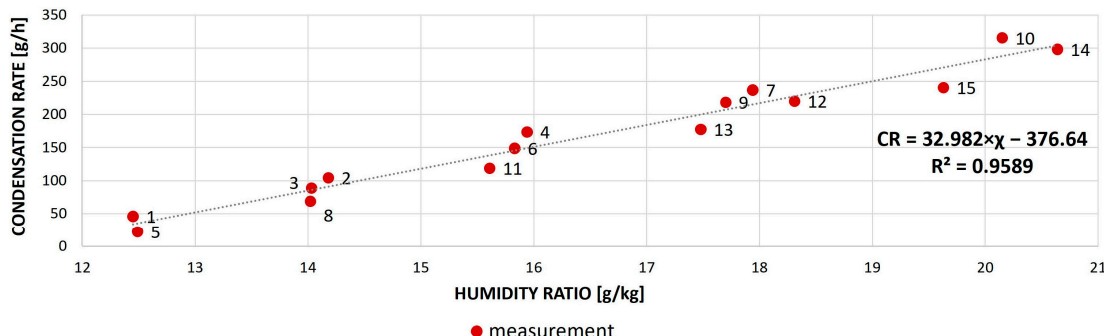

**Figure 11.** Linear dependence between humidity ratio and condensation rate.

The lowest percentage of deviation of the measured and calculated value of dehumidification performance was measurement number 14 (boundary conditions T = 30.7 °C, RH = 71.4%, $\chi$ = 20.64 g/kg), specifically 1.99%; in this case, the measured condensation rate was 298.17 g/h, and the calculated condensation rate according to Equation (2) was 304.11 g/h. The most significant difference between the measured and calculated condensation rate manifested in measurement number 5 (boundary conditions T = 25.7 °C, RH = 58.2%, $\chi$ = 12.49 g/kg). In this case, the measured condensation rate was 21.99 g/h, while using Equation (2), it was 35.31 g/h; this difference represents a percentage of deviation of 60.55%. The average percentage of deviation between all measured and calculated condensation rates was 14.24%.

### 3.2. Analysis of Results Using Nonlinear Surface Fitting

Due to fact that a high percentage of deviation between the measured and calculated condensation rate using a linear dependence analysis was discovered, for the next analysis, the dependence between condensation rate, air temperature, and relative humidity was determined using nonlinear surface fitting. The OriginPro software was chosen for this purpose. The 3D scatter plot (Figure 12a) shows the individual measurements in the spatial visualization.

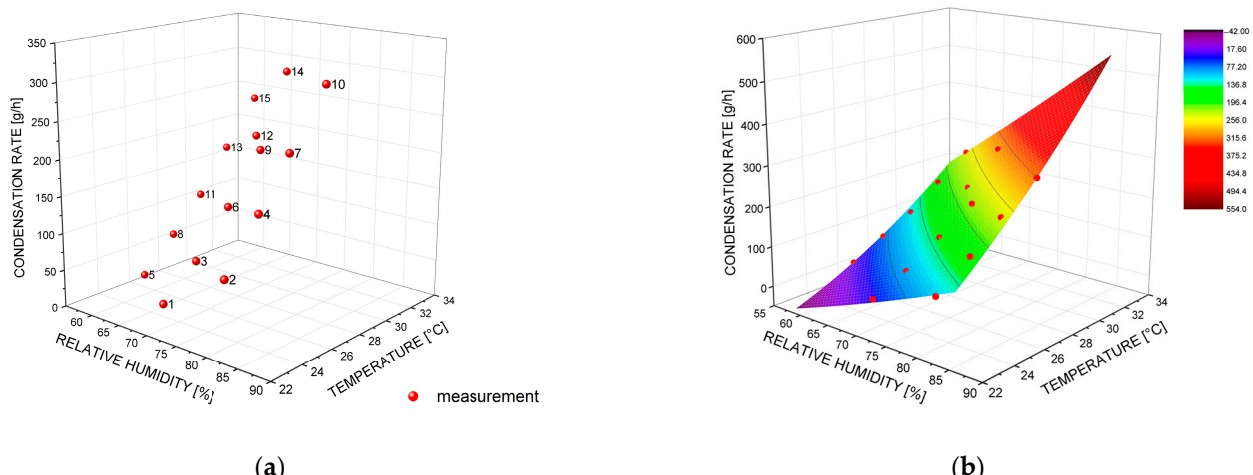

(**a**)                                      (**b**)

**Figure 12.** (**a**) 3D scatter plot—dependence between relative humidity, temperature, and condensation rate; (**b**) Nonlinear surface fitting tool predicted by OriginPro 9.0 software.

Using the two-dimensional polynomial function of the nonlinear surface fitting tool (Figure 12b), it was possible to determine the general Equation (3) of the water wall's condensation capacity as follows:

$$CR = z0 + a \times RH + b \times T + c \times RH^2 + d \times T^2 + f \times RH \times T \qquad (3)$$

where CR is the condensation rate (g/h); z0, a, b, c, d, and f are constants (z0 = 662.003, a = −11.9448, b = −61.2747, c = 0.02751, d = 0.85944, and f = 0.64519); RH is the relative humidity (%); and T is the air temperature (°C). The coefficient of determination ($R^2$) of the equation is 0.99865. Equation (3) can be used for the calculation of the water wall's condensation capacity for any relative humidity and air temperature.

In the next step, the condensation rate was calculated using Equation (3), and the percentage of deviation compared to the measured values was specified. This comparison, together with the boundary conditions, is shown in Table 5.

**Table 5.** Comparison of condensate weight measured and calculated using nonlinear surface fitting.

| | Measurement | | | | | | | | | | | | | | |
|---|---|---|---|---|---|---|---|---|---|---|---|---|---|---|---|
| | 1 | 2 | 3 | 4 | 5 | 6 | 7 | 8 | 9 | 10 | 11 | 12 | 13 | 14 | 15 |
| Temperature (°C) | 22.2 | 22.1 | 24.0 | 23.8 | 25.7 | 25.8 | 25.7 | 27.6 | 27.2 | 27.3 | 28.9 | 28.9 | 30.7 | 30.8 | 32.5 |
| Relative humidity (%) | 72.3 | 82.7 | 73.1 | 83.7 | 58.9 | 74.0 | 83.8 | 59.5 | 75.7 | 85.6 | 60.8 | 70.9 | 61.1 | 71.6 | 61.9 |
| Measurement: CR (g/h) | 46.02 | 104.29 | 88.95 | 173.42 | 21.99 | 148.81 | 236.73 | 68.81 | 218.19 | 315.36 | 118.62 | 219.92 | 177.40 | 298.17 | 240.40 |
| Equation: CR (g/h) | 42.07 | 107.88 | 92.19 | 169.27 | 23.52 | 150.71 | 237.63 | 68.35 | 214.15 | 315.90 | 118.08 | 222.98 | 174.84 | 297.67 | 241.88 |
| Absolute difference | 3.95 | 3.59 | 3.24 | 4.15 | 1.53 | 1.90 | 0.90 | 0.45 | 4.04 | 0.54 | 0.54 | 3.06 | 2.56 | 0.50 | 1.48 |
| Percent deviation (%) | 8.58 | 3.44 | 3.64 | 2.39 | 6.97 | 1.27 | 0.38 | 0.66 | 1.85 | 0.17 | 0.46 | 1.39 | 1.44 | 0.17 | 0.62 |

The lowest percentage of deviation of the measured and calculated value of dehumidification performance was found in measurement numbers 10 and 14 (boundary conditions no. 10: T = 27.3 °C, RH = 85.6%; no. 14: T = 30.8 °C, RH = 71.6%), specifically 0.17%. For measurement number 10, the measured condensation rate presented a value of 315.36 g/h, and the calculated rate according to Equation (3) was 315.90 g/h. For measurement number 14, the measured condensation rate presented a value of 298.17 g/h, and the rate calculated according to Equation (3) was 297.67 g/h. The most significant difference between the measured and calculated condensation rate manifested in measurement number 1 (boundary conditions: T = 22.2 °C and RH = 72.3%). In this case, the measured condensation rate was 46.02 g/h, while using Equation (3), it was 42.07 g/h; this difference represents a percentage of deviation of 8.58%. The average percentage of deviation between all measured and calculated condensation rates was 2.33%.

Air humidity, a significant indicator of indoor air quality, plays an important role, especially in the context of thermal comfort, where it is shown that, at the same temperature but different humidity, it is possible to feel different levels of comfort (with a high water vapor content, the environment is perceived as humid, and with a low water vapor content, as dry) [48]. The level of indoor humidity depends not only on the content of water vapor in outdoor air but also on various sources of indoor humidity and people's activities. Decorative water elements (fountains, water walls, pools) are perceived as humidifiers [1]. However, the study [49] shows that through the substitution of water with liquid desiccant, the vase or sphere with the falling film on the surface can act as a moisture receptacle that removes moisture at high humidity and releases moisture at low humidity. Thus, decorative elements can also be used to adjust humidity level.

The presented paper monitors the condensation rate on a falling water film under different boundary conditions. The values of measurements were chosen outside the area of thermal comfort (according to the ASHRAE standard [48]), especially with regard to the fact that in such a case, it is necessary to adjust the air (cooling and dehumidification) to achieve the required level of thermal comfort. Obtaining negative results using the presented Equations (2) and (3) means that dehumidification will not take place; for the respective boundary conditions, Equations (2) and (3) cannot be used (Figure 13).

| Tepmerature [°C] | 22 | 22 | 22 | 22 | 22 | 22 | 22 | 22 | 22 | 22 | 22 | 22 | 22 | 22 | 22 |
|---|---|---|---|---|---|---|---|---|---|---|---|---|---|---|---|
| Relative humidity [%] | 30 | 35 | 40 | 45 | 50 | 55 | 60 | 65 | 70 | 75 | 80 | 85 | 90 | 95 | 100 |
| Condensation rate [g/h] | no | no | no | no | no | no | no | no | 22.19 | 53.38 | 85.94 | 119.89 | 155.20 | 191.90 | 229.97 |
| Tepmerature [°C] | 23 | 23 | 23 | 23 | 23 | 23 | 23 | 23 | 23 | 23 | 23 | 23 | 23 | 23 | 23 |
| Relative humidity [%] | 30 | 35 | 40 | 45 | 50 | 55 | 60 | 65 | 70 | 75 | 80 | 85 | 90 | 95 | 100 |
| Condensation rate [g/h] | no | no | no | no | no | no | no | 11.71 | 44.75 | 79.17 | 114.96 | 152.13 | 190.67 | 230.59 | 271.89 |
| Temperature [°C] | 24 | 24 | 24 | 24 | 24 | 24 | 24 | 24 | 24 | 24 | 24 | 24 | 24 | 24 | 24 |
| Relative humidity [%] | 30 | 35 | 40 | 45 | 50 | 55 | 60 | 65 | 70 | 75 | 80 | 85 | 90 | 95 | 100 |
| Condensation rate [g/h] | no | no | no | no | no | no | no | 32.76 | 69.03 | 106.67 | 145.69 | 186.09 | 227.86 | 271.00 | 315.52 |
| Temperature [°C] | 25 | 25 | 25 | 25 | 25 | 25 | 25 | 25 | 25 | 25 | 25 | 25 | 25 | 25 | 25 |
| Relative humidity [%] | 30 | 35 | 40 | 45 | 50 | 55 | 60 | 65 | 70 | 75 | 80 | 85 | 90 | 95 | 100 |
| Condensation rate [g/h] | no | no | no | no | no | no | 17.42 | 55.54 | 95.03 | 135.90 | 178.15 | 221.77 | 266.76 | 313.13 | 360.88 |
| Temperature [°C] | 26 | 26 | 26 | 26 | 26 | 26 | 26 | 26 | 26 | 26 | 26 | 26 | 26 | 26 | 26 |
| Relative humidity [%] | 30 | 35 | 40 | 45 | 50 | 55 | 60 | 65 | 70 | 75 | 80 | 85 | 90 | 95 | 100 |
| Condensation rate [g/h] | no | no | no | no | no | no | 38.69 | 80.03 | 122.75 | 166.85 | 212.32 | 259.17 | 307.39 | 356.98 | 407.96 |
| Temperature [°C] | 27 | 27 | 27 | 27 | 27 | 27 | 27 | 27 | 27 | 27 | 27 | 27 | 27 | 27 | 27 |
| Relative humidity [%] | 30 | 35 | 40 | 45 | 50 | 55 | 60 | 65 | 70 | 75 | 80 | 85 | 90 | 95 | 100 |
| Condensation rate [g/h] | no | no | no | no | no | 18.48 | 61.67 | 106.25 | 152.19 | 199.51 | 248.21 | 298.28 | 349.73 | 402.55 | 456.75 |
| Temperature [°C] | 28 | 28 | 28 | 28 | 28 | 28 | 28 | 28 | 28 | 28 | 28 | 28 | 28 | 28 | 28 |
| Relative humidity [%] | 30 | 35 | 40 | 45 | 50 | 55 | 60 | 65 | 70 | 75 | 80 | 85 | 90 | 95 | 100 |
| Condensation rate [g/h] | no | no | no | no | no | 39.96 | 86.38 | 134.18 | 183.35 | 233.90 | 285.82 | 339.12 | 393.79 | 449.84 | 507.27 |
| Temperature [°C] | 29 | 29 | 29 | 29 | 29 | 29 | 29 | 29 | 29 | 29 | 29 | 29 | 29 | 29 | 29 |
| Relative humidity [%] | 30 | 35 | 40 | 45 | 50 | 55 | 60 | 65 | 70 | 75 | 80 | 85 | 90 | 95 | 100 |
| Condensation rate [g/h] | no | no | no | no | 14.89 | 63.16 | 112.81 | 163.83 | 216.23 | 270.00 | 325.15 | 381.67 | 439.57 | 498.85 | 559.50 |
| Temperature [°C] | 30 | 30 | 30 | 30 | 30 | 30 | 30 | 30 | 30 | 30 | 30 | 30 | 30 | 30 | 30 |
| Relative humidity [%] | 30 | 35 | 40 | 45 | 50 | 55 | 60 | 65 | 70 | 75 | 80 | 85 | 90 | 95 | 100 |
| Condensation rate [g/h] | no | no | no | no | 36.58 | 88.08 | 140.95 | 195.20 | 250.82 | 307.82 | 366.20 | 425.95 | 487.07 | 549.57 | 613.45 |
| Temperature [°C] | 31 | 31 | 31 | 31 | 31 | 31 | 31 | 31 | 31 | 31 | 31 | 31 | 31 | 31 | 31 |
| Relative humidity [%] | 30 | 35 | 40 | 45 | 50 | 55 | 60 | 65 | 70 | 75 | 80 | 85 | 90 | 95 | 100 |
| Condensation rate [g/h] | no | no | no | 6.64 | 59.99 | 114.71 | 170.81 | 228.29 | 287.14 | 347.36 | 408.96 | 471.94 | 536.29 | 602.02 | 669.12 |
| Temperature [°C] | 32 | 32 | 32 | 32 | 32 | 32 | 32 | 32 | 32 | 32 | 32 | 32 | 32 | 32 | 32 |
| Relative humidity [%] | 30 | 35 | 40 | 45 | 50 | 55 | 60 | 65 | 70 | 75 | 80 | 85 | 90 | 95 | 100 |
| Condensation rate [g/h] | no | no | no | 28.55 | 85.12 | 143.07 | 202.39 | 263.09 | 325.17 | 388.62 | 453.45 | 519.65 | 587.23 | 656.18 | 726.51 |
| Temperature [°C] | 33 | 33 | 33 | 33 | 33 | 33 | 33 | 33 | 33 | 33 | 33 | 33 | 33 | 33 | 33 |
| Relative humidity [%] | 30 | 35 | 40 | 45 | 50 | 55 | 60 | 65 | 70 | 75 | 80 | 85 | 90 | 95 | 100 |
| Condensation rate [g/h] | no | no | no | 52.17 | 111.97 | 173.14 | 235.69 | 299.62 | 364.92 | 431.60 | 499.65 | 569.08 | 639.88 | 712.06 | 785.62 |
| Temperature [°C] | 34 | 34 | 34 | 34 | 34 | 34 | 34 | 34 | 34 | 34 | 34 | 34 | 34 | 34 | 34 |
| Relative humidity [%] | 30 | 35 | 40 | 45 | 50 | 55 | 60 | 65 | 70 | 75 | 80 | 85 | 90 | 95 | 100 |
| Condensation rate [g/h] | no | no | 15.86 | 77.51 | 140.54 | 204.94 | 270.71 | 337.87 | 406.39 | 476.30 | 547.57 | 620.23 | 694.26 | 769.66 | 846.44 |
| Temperature [°C] | 35 | 35 | 35 | 35 | 35 | 35 | 35 | 35 | 35 | 35 | 35 | 35 | 35 | 35 | 35 |
| Relative humidity [%] | 30 | 35 | 40 | 45 | 50 | 55 | 60 | 65 | 70 | 75 | 80 | 85 | 90 | 95 | 100 |
| Condensation rate [g/h] | no | no | 39.69 | 104.57 | 170.82 | 238.45 | 307.45 | 377.83 | 449.58 | 522.71 | 597.22 | 673.10 | 750.35 | 828.98 | 908.99 |

no working  CR ≤ 100 g/h  100 < CR ≤ 200 g/h  200 < CR ≤ 300 g/h

300 < CR ≤ 400 g/h  400 < CR ≤ 500 g/h  CR > 500 g/h

**Figure 13.** Condensation rate on a falling water film with an area of 1 m$^2$ for the selected temperature and relative humidity using Equation (3).

Currently, various air-conditioning systems are most often used to achieve required humidity levels. Condensate dehumidifiers and desiccant dehumidifiers are used most often [50]. On the market, it is possible to find several commercial systems, from the smallest designed for small rooms, whose dehumidification capacity is about 0.3 L per 24 h, through medium mobile dehumidifiers designed for individual rooms at home with an average dehumidification capacity of about 8–30 L per 24 h (at a temperature of 30 °C and relative humidity of 80%) and to large complex air-handling units with a built-in system for adjusting temperature and humidity, which are suitable for buildings with controlled ventilation. At an air temperature of 30 °C and relative humidity of 80%, the dehumidification capacity of a water wall with an effective area of 1 m$^2$, according to Equation (3), is 366.2 g/h. This is approximately 8.8 L per 24 h, and this dehumidification capacity is comparable to that of a medium mobile dehumidifier.

Condensation on thin falling liquid films that fall on solid surfaces is a common and very important process for technical applications and is wildly used in air-conditioning systems [37]. The results of this paper show that this process can also be used in the case of water walls with a controlled water temperature. Conventional air-conditioning systems are closed systems with controlled inlet air velocity and direction, compared to water walls where dehumidification occurs directly in the room where they are placed and without airflow control.

## 4. Conclusions

This paper shows the possibility of the use of water walls as dehumidifiers, although this element is built in interior spaces mainly for its aesthetic function. The subject of the research was a water wall with controlled water temperature and effective area of falling water film of 1 m$^2$. The main conclusions are as follows:

- Testing in laboratory conditions in the climate chamber allowed the definition of water walls with controlled water temperature as a device with a certain level of performance. The measurement of increments of condensate using the laboratory scale ensured the accurate definition of changes in time.
- The results showed that dehumidification capacity is a dynamic process that strongly depends on the boundary conditions of the air.
- The water wall dehumidification capacity varied in the range of 21.99–315.36 g/h for the tested measurements. This represents a difference between the lowest and highest performance of 293.37 g/h.
- Equation (3) determined by nonlinear surface fitting shows a higher coefficient of determination (R$^2$ = 0.99865) and a lower average percentage of deviation between measured and calculated values (2.33%) compared to Equation (2) generated by linear dependence (R$^2$ = 0.9589; average percentage of deviation = 14.24%).
- The achieved results expand our knowledge regarding the use of water elements to adjust the parameters of indoor air and their ability to connect with convectional air treatment systems with the aim to reduce energy consumption.

A water wall with a controlled water temperature was tested in laboratory conditions in a climate chamber in order to determine its dehumidification capacity regardless of the surrounding materials or room volume. However, the real effects of water walls in interior spaces will depend on the sorption capacity of the elements and the surrounding materials in the interior of buildings. Because of this, it is necessary to develop this topic and to examine the dehumidification capacity of water walls in real conditions of sorption-active surfaces.

**Author Contributions:** Conceptualization, K.C. and F.V.; methodology, K.C.; software, M.V.; validation, K.C., F.V. and M.V.; formal analysis, K.C.; investigation, K.C.; resources, K.C.; data curation, K.C.; writing—original draft preparation, K.C.; writing—review and editing, Z.V. and M.V.; visualization, K.C.; supervision, Z.V.; project administration, Z.V.; funding acquisition, Z.V. All authors have read and agreed to the published version of the manuscript.

**Funding:** This research received no external funding.

**Institutional Review Board Statement:** Not applicable.

**Informed Consent Statement:** Not applicable.

**Data Availability Statement:** All data are presented in this article in the form of figures and tables.

**Acknowledgments:** The authors are extremely grateful to the support in this undertaking of the Ministry for Education of the Slovak Republic with VEGA 1/0217/19 "Research of Hybrid Blue and Green Infrastructure as Active Elements of a Sponge City", a project run by the Slovak Research and Development Agency APVV-18-0360 "Active hybrid infrastructure closer to a sponge city" and a project run by the Operational Programme Research, Development and Education of The Ministry of Education, Youth and Sports of the Czech Republic CZ.02.2.69/0.0/0.0/18_053/0016962 "International mobility of researchers at the Brno University of Technology II".

**Conflicts of Interest:** The authors declare no conflict of interest.

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
