# Peer review of "Determination of Dehumidification Capacity of Water Wall with Controlled Water Temperature: Experimental Verification under Laboratory Conditions"

_sustainability, doi:10.3390/su13105684_

Round 1

Reviewer 1 Report

This is an interesting experimental work with a water wall installed inside a climate chamber, to study dehumidification capacity of the water wall.

The following points should be addressed in a revised version by the authors:

Based on the sensors’ accuracy specifications stated in Table 1, it is not reasonable to give temperature and humidity values to two decimal digits. Please delete the second decimal digit.

Figure 10 increase fonts

Lines 303-303 correct to “correlation coefficient”

Please replace “specific humidity” with “humidity ratio” or “absolute humidity”

Figure 11: It is known from mass transfer theory, that the condensation rate is proportional to the humidity gradient. However,  

The authors should observe in which points they have dropwise condensation mode, or film mode (laminar or turbulent). Then they should use what is well known from the theory of condensation for approximate calculation of condensation rate, (see for example the well-known heat and mass transfer textbook: Incropera and deWitt, Wiley, 2017), to depict their measurement points on a psychrometric chart, calculate the Nusselt number, condensation rate and cooling rate required in their experiment and compare to the measurement results and eq. (3).   

Lines 366-378 Dehumidification by use of a vertical water wall would require a big surface area, because the temperature difference is small to produce the required cooling rate for the condensation. This should be discussed to avoid misunderstanding.  

Author Response

Response to Reviewer 1 Comments

see in the attachment

Reviewer 2 Report

The paper includes an experimental assessment of the dehumidification capacity of water wall. The topic is interesting and in line with the editorial directions of the journal. The paper is well written and can be published after the following changes will be made:

  • Figure 1: please cite references for all the images not owned by the authors.
  • In the results, since temperature and humidity are correlated, it is difficult to read the results and understand the benefits of the water wall. It would be good to present the results in terms using the psychrometric chart. 
  • From the analysis of figure 12, it seems that the CR is more influenced by Temperature than by relative humidity. Any physical explanation of this phenomenon?
  • In general, I am not totally convinced of the equations presented. For example, if we put very low values of RH and high values of T (e.g. hot arid conditions), still the dehumidification process happens, which is a bit strange under the physical point of view. Also, negative results can be obtained for some RH-T couples, which are still in the range of possible values. At least some boundary conditions of the formula should be presented.
  • A proofread by an English native speaker is needed. There are several sentences not totally clear.

Author Response

Response to Reviewer 2 Comments

please see in the attachment

Reviewer 3 Report

The manuscript deals with a concept of using a water wall as an air dehumidification system. The paper itself is interesting and it certainly has a large practical application potential (due to the use of such systems in buildings). I would recommend to publish the paper after the Authors have made adequate improvements to the manuscript according to the comments below: 
-The introduction provides a background for the investigation and is quite interesting. However, I would expect the Authors to provide more references (preferably journal papers) that deal with the exact same problem of air dehumidification with the use of water systems (how other authors dealt with this issue, what their results and conclusions were and etc. ). Besides, at the end of the introduction, I would expect the Authors to provide clearly the novelty of their paper.
-The most crucial part of the paper is an experimental study of the condensation process of water vapour from the air (of various temperature and relative humidity values). It would be most interesting to compare the “actual” experimental data obtained in the current study with theoretical values determined from the mass transfer analysis – Has the Authors made an effort in this area? If not, a short discussion on this issue (e.g. based on works by other authors) might be welcome to be added as a separate paragraph.  
-In line 247 it is mentioned that “average values” are evaluated. Can the Authors provide data (e.g. as supplementary material or in a table within the manuscript) on standard deviations and other statistically important elements?  
Other comments:
-There should be no text directly below title of chapter 2 and above subchapter 2.1. If subchapters are used, all the text should be there e.g. in 2.1, 2.2 and etc. Thus, I suggest to add another subchapter (before 2.1) to accommodate the text in the lines 108 – 122.
-Figure 1 should be placed closer to where it is mentioned in the text.
-Line 568: ref. 52: shouldn’t it be “ASHRAE”?
-The English language is understood, however grammar mistakes are present throughout the paper and the proofreading of the manuscript by a professional translator might be necessary; examples of errors: in lines 42-42 instead of “In this case water, water falls on solid 43 material” it should probably be “In this case water falls on solid 43 material”; in line 56 instead of “this issue was not detailed explored” should be e.g. “this issue was not explored in detail”, again in line 56 instead of “It does not exist a clear…” should be e.g. “There is no clear…”, in line 68 instead of “air temperature decrease” should be “air temperature decreases” and others. 

Author Response

Response to Reviewer 3 Comments

please see in the attachment

Round 2

Reviewer 1 Report

Yes it cn be published now

Reviewer 3 Report

The paper has been improved and my comments have been addressed and considered. I can now recommend the paper to be published.